# enChIP-Seq Analyzer: A Software Program to Analyze and Interpret enChIP-Seq Data for the Detection of Physical Interactions between Genomic Regions

**DOI:** 10.3390/genes13030472

**Published:** 2022-03-07

**Authors:** Ashita Sarudate, Toshitsugu Fujita, Takahiro Nakayama, Hodaka Fujii

**Affiliations:** 1Research Institute of Bio-System Informatics, Tohoku Chemical Co., Ltd., 6-15-5 Mitake, Morioka 020-0122, Iwate, Japan; sarudate@t-kagaku.co.jp (A.S.); nakayama@t-kagaku.co.jp (T.N.); 2Department of Biochemistry and Genome Biology, Hirosaki University Graduate School of Medicine, 5 Zaifu-cho, Hirosaki 036-8562, Aomori, Japan; toshitsugu.fujita@hirosaki-u.ac.jp

**Keywords:** enChIP-Seq, CRISPR, intergenomic interactions, 3-D genomics, enChIP-Seq analyzer

## Abstract

Accumulating evidence suggests that the physical interactions between genomic regions play critical roles in the regulation of genome functions, such as transcription and epigenetic regulation. Various methods to detect the physical interactions between genomic regions have been developed. We recently developed a method to search for genomic regions interacting with a locus of interest in a non-biased manner that combines pull-down of the locus using engineered DNA-binding molecule-mediated chromatin immunoprecipitation (enChIP) and next-generation sequencing (NGS) analysis (enChIP-Seq). The clustered regularly interspaced short palindromic repeats (CRISPR) system, consisting of a nuclease-dead form of Cas9 (dCas9) and a guide RNA (gRNA), or transcription activator-like (TAL) proteins, can be used for enChIP. In enChIP-Seq, it is necessary to compare multiple datasets of enChIP-Seq data to unambiguously detect specific interactions. However, it is not always easy to analyze enChIP-Seq datasets to subtract non-specific interactions or identify common interactions. To facilitate such analysis, we developed the enChIP-Seq analyzer software. It enables easy extraction of common signals as well as subtraction of non-specific signals observed in negative control samples, thereby streamlining extraction of specific enChIP-Seq signals. enChIP-Seq analyzer will help users analyze enChIP-Seq data and identify physical interactions between genomic regions.

## 1. Introduction

The dynamics of the three-dimensional structure of the genome play a critical role in the regulation of genome functions such as transcription, epigenetic regulation, genomic imprinting, and X-chromosome inactivation [1,2]. Methods to detect physical interactions between genomic regions have been developed in the last two decades. Chromosome conformation capture (3C) was the first biochemical method widely used to detect physical interactions between genomic regions [3]. Various derivatives of 3C have been developed [4]. 3C and its derivatives depend on cross-linking, fragmentation of chromatin by enzymatic digestion or other methods, and ligation of proximal DNA ends. The ligation step can be a source of artifactual “interactions”, leading to occasional discrepancies relative to results derived from other methods, such as imaging analysis [5,6]. Therefore, several “ligation-free” biochemical methods were developed to detect physical interactions between genomic regions. Genome architecture mapping (GAM) is a method to measure the statistical proximities of genomic regions by sequencing the DNA extracted from ultra-thin cryosectioned nuclear slices [7]. Split-pool recognition of interactions by tag extension (SPRITE) is a method that involves performing repeated rounds of splitting and barcoding of individual chromatin complexes followed by identification of the interacting genomic regions by matching the barcodes [8]. Chromatin interaction analysis via droplet-based and barcode-linked sequencing (ChIA-Drop) tracks amplicons arising from gel-bead-in-emulsion (GEM) droplets of each chromatin complex by barcode sequencing [9].

We recently developed a method to search for genomic regions interacting with a locus of interest in a non-biased manner that combines pull-down of the locus using engineered DNA-binding molecule-mediated chromatin immunoprecipitation (enChIP) and next-generation sequencing (NGS) analysis (enChIP-Seq). The CRISPR system, consisting of a nuclease-dead form of Cas9 (dCas9) and a guide RNA (gRNA), or transcription activator-like (TAL) proteins, can be used for enChIP. In enChIP-Seq, it is necessary to compare multiple datasets of enChIP-Seq data to unambiguously detect specific interactions. However, it is not always easy to analyze enChIP-Seq datasets to subtract non-specific interactions or identify common interactions. To facilitate such analysis, we developed the enChIP-Seq analyzer software. It enables easy extraction of common signals as well as subtraction of non-specific signals observed in negative control samples, thereby streamlining extraction of specific enChIP-Seq signals. enChIP-Seq analyzer will help users analyze enChIP-Seq data and identify physical interactions between genomic regions.

## 2. Materials and Methods

### 2.1. Implementation

enChIP-Seq analyzer is provided as free software at GitHub (https://github.com/TKY-SE/enChIP-Seq-Analyzer, accessed on 1 December 2021). All the program was executed on a computer with Intel^®^ Core™ i5-7500 CPU @ 3.40 GHz 3.41 GHz, and 8 GB of RAM. Preparation of data sets, mode of analysis, and system handling for enChIP-Seq analyzer are shown in a step-by-step manner below.

### 2.2. Procedures for enChIP-Seq

The procedures to use enChIP-Seq to detect genomic regions interacting with a locus of interest depend on how the locus is tagged with an engineered DNA-binding molecule [10,11]. In “in cell” enChIP-Seq, an engineered DNA-binding molecule, such as a CRISPR complex, is expressed in the cells to be analyzed. After cross-linking with formaldehyde or other cross-linkers, if necessary, chromatin is fragmented using sonication or enzymatic digestion. Subsequently, the tagged locus is isolated using affinity purification, and the interacting genomic regions are identified by NGS [10]. In in vitro enChIP-Seq, chromatin is cross-linked, if necessary, fragmented, and then incubated with the CRISPR complex, consisting of a recombinant dCas9 protein and a synthetic or in vitro transcribed gRNA, for in vitro locus tagging. Subsequently, the tagged locus is isolated using affinity purification, and the interacting genomic regions are identified by NGS [11].

The CRISPR complex also binds to off-target sites in addition to the target genomic regions [12,13,14,15]. Therefore, to identify genomic regions truly interacting with the target site, it is necessary to analyze negative control samples, such as cells expressing only dCas9 or dCas9 plus irrelevant gRNAs. In addition, multiple gRNAs for the target locus should be analyzed to extract common peaks from the NGS analysis, which can then be identified as true positives with higher confidence. Consequently, it is necessary to compare multiple negative controls and multiple samples with on-target gRNAs. It is tedious to compare these samples manually, and this observation prompted us to develop a software program to automate comparison of multiple NGS peaks.

### 2.3. Filtering of the NGS Peaks to Identify Genomic Regions Interacting with a Target Locus

enChIP-Seq analyzer is a software program that enables easy consolidation of enChIP-Seq data, including extraction of common peaks observed for different gRNAs bound to the target genomic region and subtraction of peaks in negative control samples (Figure 1).

### 2.4. Preparation of the Data Set

To extract enChIP-specific NGS peaks, Model-based Analysis of ChIP-Seq (MACS) [16] is used to compare NGS data from enChIP with that for input DNA. MACS data for enChIP-specific NGS peaks is exported as a tab file (Figure 2), which is used by enChIP-Seq analyzer.

### 2.5. Mode of Analysis

enChIP-Seq analyzer extracts common peak information or eliminates negative peak information, as shown in Figure 3.

Extraction of common peak information. When two or more tab files are analyzed, overlapped peak regions (red double-headed arrow) are extracted as common peak information (region).

Elimination of negative peak information. When two or more tab files are analyzed, union range regions (green double-headed arrow) are eliminated as negative peak information (region).

### 2.6. Handling of the System

enChIP-Seq analyzer, a Java-based software program, is designed for use with Microsoft Windows 10. The software can be downloaded from GitHub (https://github.com/TKY-SE/enChIP-Seq-Analyzer, accessed on 1 December 2021). The browser for the software is shown in Figure 4A. The detailed procedures are outlined below.

#### 2.6.1. Extraction of Peak Information Commonly Included in Two or More Tab Files

(1-1) Add tab files that include enChIP-specific NGS peaks in the field “I”. To this end, click on “add” to open a selection window and select a tab file from your computer. Repeat this step to add additional tab files.

(1-2) (If necessary) To remove an added file, select a file(s) and click on “remove”.

(1-3) (Mandatory) Select an added file as a base file, which is then highlighted.

(1-4) To execute the program, click on “view”.

(1-5) The result is shown in “V”, which is based on information from the base file. The number of common peaks is shown in “VI” as “count”. An example is shown in Figure 4B.

#### 2.6.2. Elimination of Negative Peak Information 

(2-1) Perform steps (1-1) through (1-3). For this analysis, use of only one tab file is also acceptable.

(2-2) Add a tab file(s) that includes NGS peaks with negative peak information (e.g., NGS peaks from enChIP experiments performed without a gRNA) in the field “II”. To this end, click on “add” to open a selection window and select a tab file(s) from your computer. Repeat this step to add two or more tab files.

(2-3) (If necessary) To remove an added file, select a file(s) and click on “remove” in “II”.

(2-4) (Mandatory) Select an added file in “I” as a base file, which is then highlighted.

(2-5) To execute the program, click on “Pros.-Neg. (IV)”.

(2-6) The result is shown in “V”, which is based on the information from the base file. The number of peaks remaining after elimination of negative peak information is shown in “VI” as “count”. An example is shown in Figure 4C.

(2-7) (If necessary) If tag number and/or fold enrichment are necessary criteria for filtering, set tag number and/or fold enrichment criteria in “III”. The following symbols can be used for filtering: ≥, =, and ≤.

#### 2.6.3. Data Export

Data shown in “V” can be exported as a csv file and easily utilized by other software/web tools. To this end, click on “Export Table (VII)”. Select a folder to save the file from the opened window.

#### 2.6.4. View Information within a Tab File

(1)The information within a tab file can be viewed directly in the software program. To this end, add one tab file that includes enChIP-specific NGS peaks in the field “I” (or “II”).(2)Click on “view” in the field “I” (or “II”) to view the information within the file.(3)The information is shown in “V”.

## 3. Results and Discussion

We previously identified genomic regions associated with the 5′HS5 locus on a genome-wide scale in K562 cells under undifferentiated or sodium butyrate (NaB)-mediated differentiated conditions [10]. In that study, we performed enChIP using gRNAs #6 and #17, which targeted the 5′HS5 locus and used the purified DNA for NGS analysis. As a negative control, we also performed enChIP in the absence of a gRNA and used the purified DNA for NGS analysis. Next, we compared the NGS peaks from each enChIP experiment to those for input DNA and used MACS analysis to extract the list of enriched peaks (“enChIP #6 peaks”, “enChIP #17 peaks”, and “Off-target sites”) (Data sets in Figure 5A). Using enChIP-Seq analyzer, we compared the lists of extracted peaks in a step-by-step manner as follows (Figure 5A).
Step 1:
(1)We compared the data from “enChIP #6 peaks” and “Off-target sites” to eliminate off-target binding sites. The resultant information was named “enChIP #6-specific sites”.(2)We compared the data from “enChIP #17 peaks” and “Off-target sites” to eliminate off-target binding sites. The resultant information was named “enChIP #17-specific sites”.(3)To identify peaks with confidence, we adopted two criteria for choosing peaks based on NGS information from the target 5′HS5 locus: (i) Tag number ≥5% of that of the target 5′HS5 locus (which can be considered as an interacting ratio of ≥5%) and (ii) fold enrichment relative to input genomic DNA ≥10. In this regard, 19 and 228 peaks for “enChIP #6-specific sites” and “enChIP #17-specific sites”, respectively, passed the two criteria.Step 2:
We compared the data from “enChIP #6-specific sites” and “enChIP #17-specific sites” to extract “enChIP #6/#17-common sites”. We extracted six peaks, which can be considered *bona fide* physically interacting genomic regions (Figure 5B). These results were consistent with those from our previous study [10] (please note that in Ref. [10], the symbols > should have been ≥).

## 4. Conclusions

In this study, we developed enChIP-Seq analyzer to compare multiple enChIP-Seq datasets to unambiguously detect specific interactions. enChIP-Seq analyzer is simple and easy to use. We believe that enChIP-Seq analyzer will help users to analyze enChIP-Seq data and detect physical interactions between genomic regions. In addition, the software can be used to compare ChIP-Seq datasets to extract common peaks.

## 5. Availability and Requirements

Project name: enChIP-Seq analyzerSoftware homepage (GitHub): https://github.com/TKY-SE/enChIP-Seq-Analyzer, accessed on 1 December 2021Programming language: JavaOther requirements: Windows machineLicense: NoneAny restrictions to use by non-academics: None.Notes: We have a plan to adapt the software for Linux and Mac. As soon as they are available, we will upload the code in GitHub

## Figures and Tables

**Figure 1 genes-13-00472-f001:**
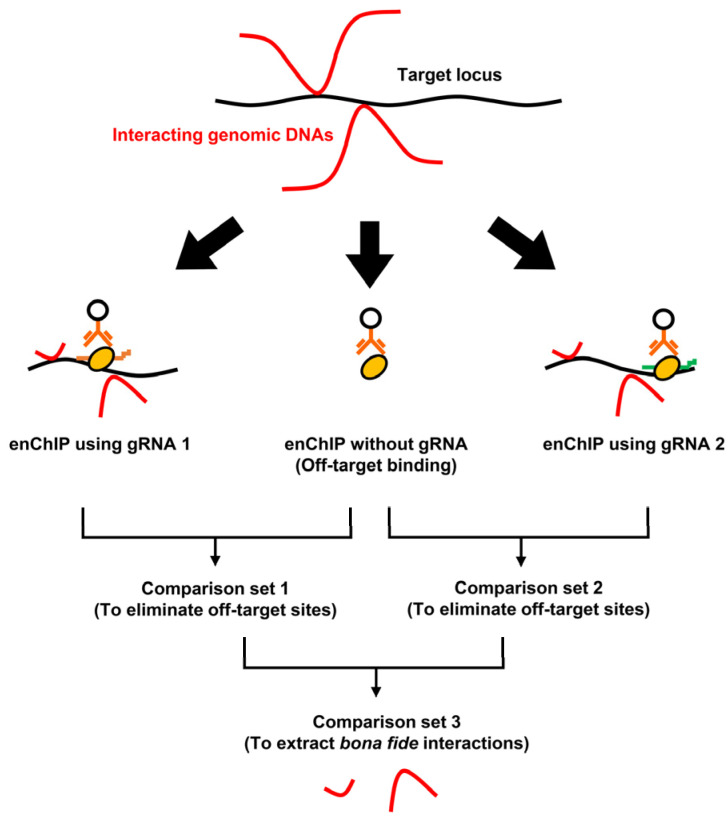
An example of using enChIP-Seq to identify interacting genomic regions. enChIP-Seq was performed in the presence of different gRNAs or in the absence of a gRNA (a negative control). Peaks detected by enChIP without gRNA were subtracted from those for enChIP-Seq with gRNAs to eliminate off-target sites (Comparison sets 1 and 2). Subsequently, the resultant data could be compared to extract *bona fide* interactions.

**Figure 2 genes-13-00472-f002:**
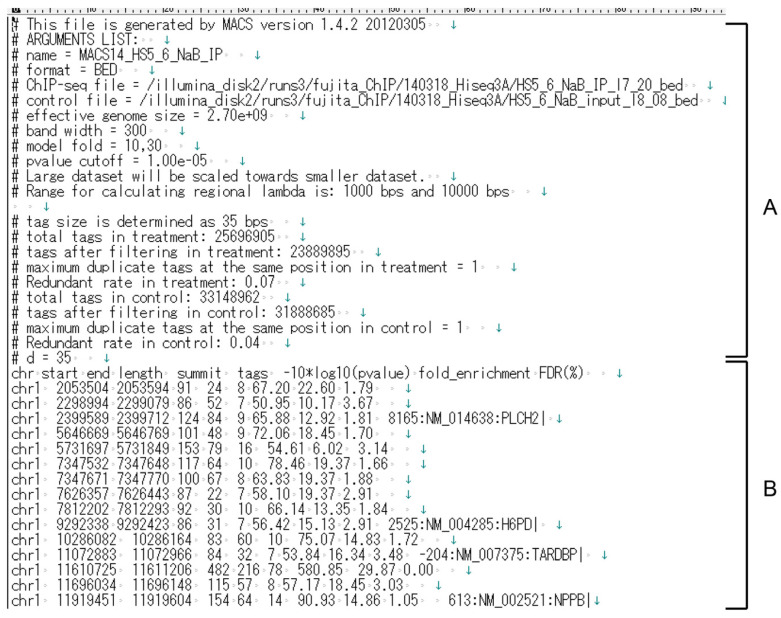
An example of a tab file used in enChIP-Seq analyzer. (**A**) Lines starting with # are comments and are not processed by the software. (**B**) Lines separated by tags are used for calculations in enChIP-Seq analyzer.

**Figure 3 genes-13-00472-f003:**
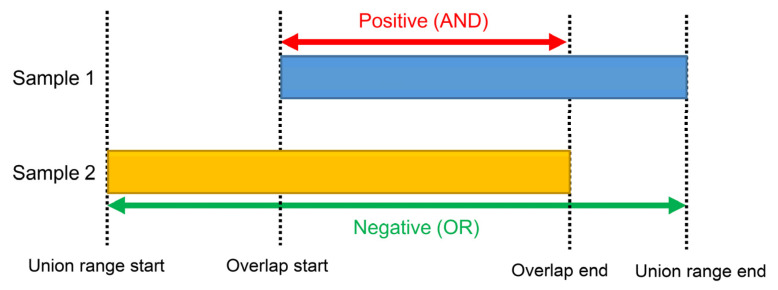
Mode of analysis. enChIP-Seq analyzer extracts overlapped regions (Positive (AND)) as common peak information or eliminates union range regions (Negative (OR)) as negative peak information.

**Figure 4 genes-13-00472-f004:**
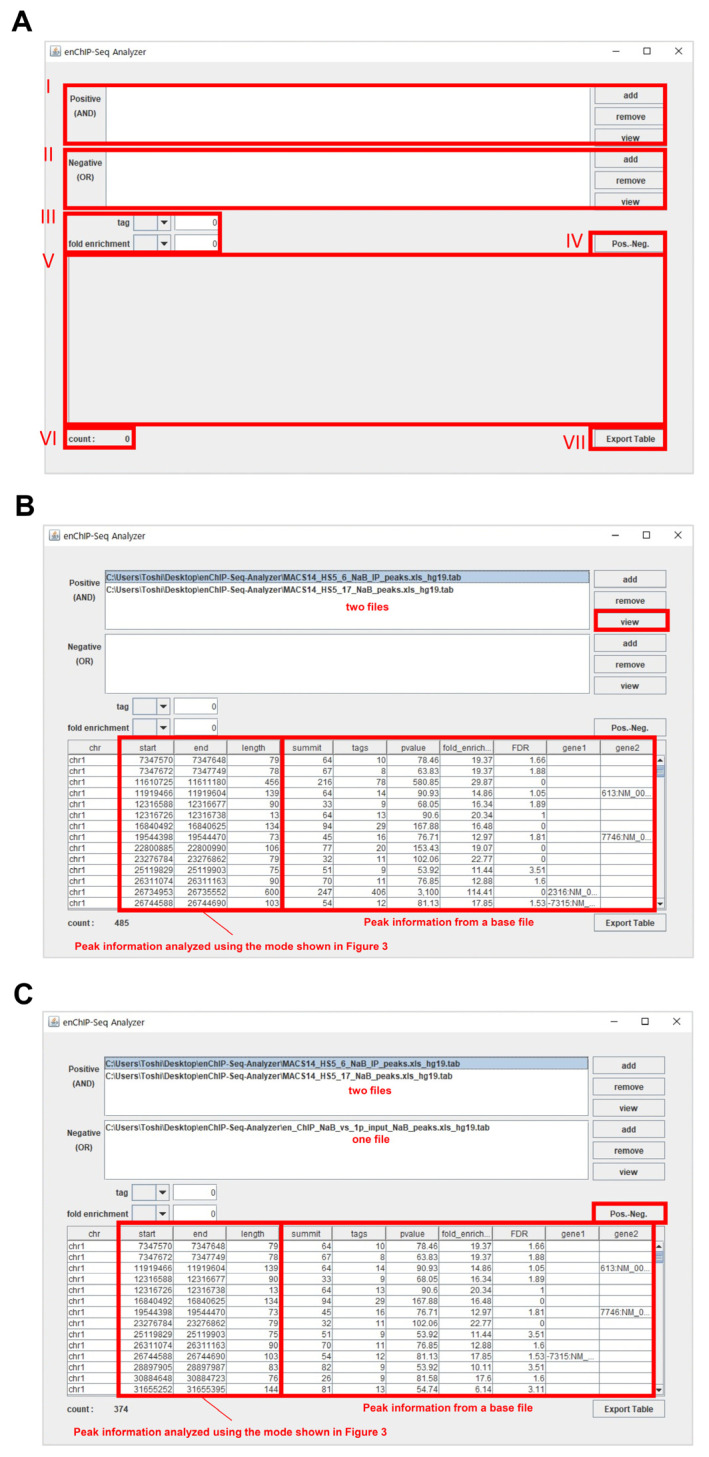
The main screen of enChIP-Seq analyzer. (**A**–**C**) Examples of the screens of enChIP-Seq analyzer. Handling processes are shown in the main text.

**Figure 5 genes-13-00472-f005:**
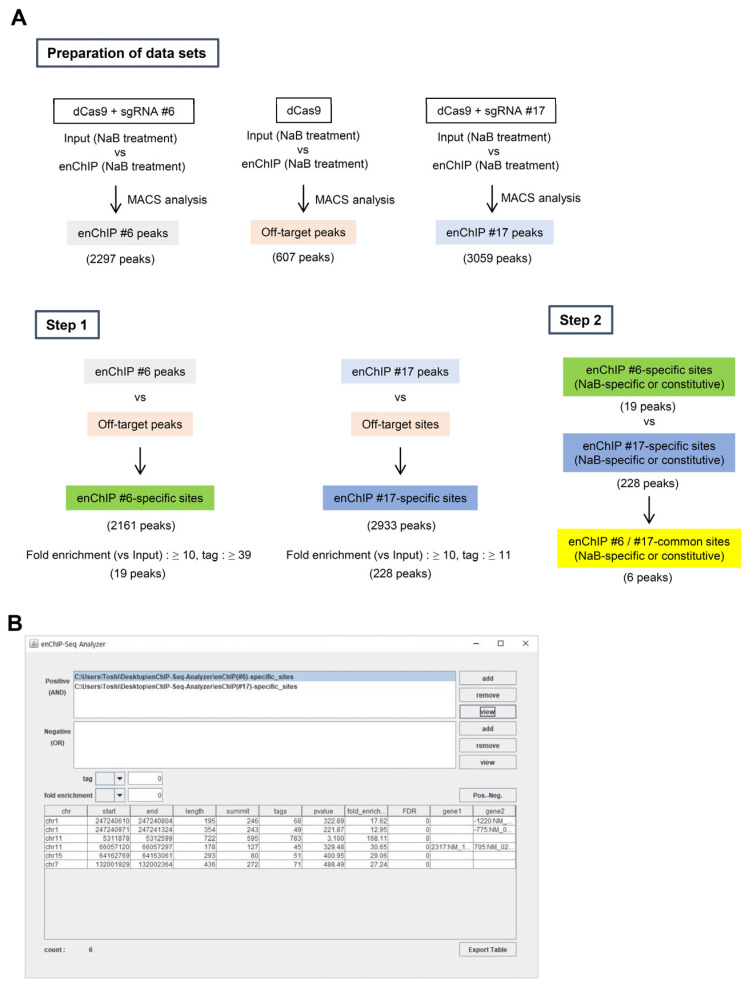
An example of results from enChIP-Seq analyzer. (**A**) Step-by step procedures to extract *bona fide* genomic regions interacting with a target genomic region. (**B**) An example of the results from enChIP-Seq analyzer.

## Data Availability

The software is available at: https://github.com/TKY-SE/enChIP-Seq-Analyzer, accessed on 1 December 2021. The data is publicly available and described at our previous paper [10].

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
