# Peer review of "enChIP-Seq Analyzer: A Software Program to Analyze and Interpret enChIP-Seq Data for the Detection of Physical Interactions between Genomic Regions"

_genes, 2022, doi:10.3390/genes13030472_

Round 1

Reviewer 1 Report

The authors have developed a java-based  enChIP-Seq analyzer to compare multiple enChIP-Seq

datasets to unambiguously detect specific interactions. The software was designed to detect physical interactions between genomic regions and compare datasets with common peaks. It is a useful toolkit for general readers in bio-informatics and bio-engineering. Here comes some comments which may help to improve the manuscript.

  1. Figure 5 B needs to be improve in quality. It seems not clear. The same issues are with Figure 4.

2.Whether it can be compared and analyzed with the same type of software to find the differences and accuracy of common peaks?

Author Response

The authors have developed a java-based enChIP-Seq analyzer to compare multiple enChIP-Seq datasets to unambiguously detect specific interactions. The software was designed to detect physical interactions between genomic regions and compare datasets with common peaks. It is a useful toolkit for general readers in bio-informatics and bio-engineering.

We thank the reviewer for their encouraging comment.

  1. Figure 5 B needs to be improved in quality. It seems not clear. The same issues are with Figure 4.

We thank the reviewer for their comments. We improved the resolution of the figures so that text in the figures is now easily legible.

  1. Whether it can be compared and analyzed with the same type of software to find the differences and accuracy of common peaks?

To the best of our knowledge, there is no same type of software. In fact, that is the main reason why we attempted to generate a software to achieve our goal.

Reviewer 2 Report

Gist: The others develop a need of the hour software called enChIP-Seq analyzer which compares  multiple ChIP-Seq datasets and identifies interactions. The software is java based, user friendly and is employed for detecting physical interactions between genomic regions. The software also uses extraction of common peaks for analysis. 

Strengths: The manuscript is generally written well with software description neatly tabulated for laymen.

Weaknesses/Limitations;  The authors could perhaps use some other test cases and in their software they could give those examples for naive users to assess the datasets

A word on reinstantiability of codes could be mentioned 

The figure 1 legend is bit tilted and could be rewritten:  What do authors mean by  "Data for enChIP-Seq with gRNAs was compared with that without a gRNA to eliminate off-target sites"  It is not clear

Sections 2.6.1 and 2.6.2 could be subtly considered as supplementary information.  Not sure, this appears more like a protocol. 

I recommend the authors use jar files and put the code/software in Github.  Wouldn't enchip analyzer work for Linux?  As it is Java based, I tried using bash/linux command line, it works. A sentence or two may be mentioned for linux or Mac users in "Notes" section. 

I ran the software myself and crosschecked it.

The Figure 5 could perhaps come first 

Scores on a scal eof 0-5 with 5 being the best

Language: 3

Novelty: 4

Scope/Importance: 4

brevity: 3.5

Author Response

Gist: The others develop a need of the hour software called enChIP-Seq analyzer which compares multiple ChIP-Seq datasets and identifies interactions. The software is java based, user friendly and is employed for detecting physical interactions between genomic regions. The software also uses extraction of common peaks for analysis.

Strengths: The manuscript is generally written well with software description neatly tabulated for laymen.

Weaknesses/Limitations; The authors could perhaps use some other test cases and in their software they could give those examples for naive users to assess the datasets

We thank the reviewer for appreciating the utility of the software.

  1. A word on reinstantiability of codes could be mentioned.

We may misunderstand the meaning of "reinstantiability", but if it means whether the enChIP-Seq analyzer software exports data outputs in a machine-readable format, it outputs CVS files. Therefore, output data can be utilized by other software / web tools easily. We added this info in the revised manuscript (2.6.3. Data export). If our interpretation is wrong, it would be great if you could kindly elaborate the meaning of "reinstantiability".

  1. The figure 1 legend is bit tilted and could be rewritten: What do authors mean by "Data for enChIP-Seq with gRNAs was compared with that without a gRNA to eliminate off-target sites" It is not clear.

According to the reviewer’s comment, we revised the legend of Figure 1. We believe that it is now clear.

  1. Sections 2.6.1 and 2.6.2 could be subtly considered as supplementary information. Not sure, this appears more like a protocol.

We appreciate the comment by the reviewer. We agree that the sections might be those for Supplementary Information. However, we also think that moving them to Supplementary Information would be inconvenient for readers because they need to read different files. Considering that these sections are short, we keep them as they are and think that would not do harm.

  1. I recommend the authors use jar files and put the code/software in Github. Wouldn't enchip analyzer work for Linux? As it is Java based, I tried using bash/linux command line, it works. A sentence or two may be mentioned for linux or Mac users in "Notes" section.

We thank the reviewer for their insightful comment. We have put the code in Github, and its link info is now described in the main text (5. Availability and requirements and Data Availability Statement).

In addition, we have a plan to adapt the software for Linux and Mac. We added the comment in the 'Notes' section.

  1. I ran the software myself and crosschecked it.

We appreciate kindness of the reviewer.

  1. The Figure 5 could perhaps come first.

Thank you for the suggestion. Either could work. Since we prefer explaining the principle first and moving to an example, you may understand that we keep the order of the figures as they are to avoid drastic change in the structure of the manuscript.